# Evolution of Resistance to Irinotecan in Cancer Cells Involves Generation of Topoisomerase-Guided Mutations in Non-Coding Genome That Reduce the Chances of DNA Breaks

**DOI:** 10.3390/ijms24108717

**Published:** 2023-05-13

**Authors:** Santosh Kumar, Valid Gahramanov, Shivani Patel, Julia Yaglom, Lukasz Kaczmarczyk, Ivan A. Alexandrov, Gabi Gerlitz, Mali Salmon-Divon, Michael Y. Sherman

**Affiliations:** 1Department of Molecular Biology, Faculty of Natural Sciences, Ariel University, Ariel 40700, Israel; shashisantosh2007@gmail.com (S.K.);; 2Department of Anatomy and Anthropology & Department of Human Molecular Genetics and Biochemistry, Sackler Faculty of Medicine, Tel Aviv University, Tel Aviv 6997801, Israel; 3Adelson School of Medicine, Ariel University, Ariel 40700, Israel

**Keywords:** SN38 resistance, mutations, non-coding genome, DNA repair, dose escalation

## Abstract

Resistance to chemotherapy is a leading cause of treatment failure. Drug resistance mechanisms involve mutations in specific proteins or changes in their expression levels. It is commonly understood that resistance mutations happen randomly prior to treatment and are selected during the treatment. However, the selection of drug-resistant mutants in culture could be achieved by multiple drug exposures of cloned genetically identical cells and thus cannot result from the selection of pre-existent mutations. Accordingly, adaptation must involve the generation of mutations de novo upon drug treatment. Here we explored the origin of resistance mutations to a widely used Top1 inhibitor, irinotecan, which triggers DNA breaks, causing cytotoxicity. The resistance mechanism involved the gradual accumulation of recurrent mutations in non-coding regions of DNA at Top1-cleavage sites. Surprisingly, cancer cells had a higher number of such sites than the reference genome, which may define their increased sensitivity to irinotecan. Homologous recombination repairs of DNA double-strand breaks at these sites following initial drug exposures gradually reverted cleavage-sensitive “cancer” sequences back to cleavage-resistant “normal” sequences. These mutations reduced the generation of DNA breaks upon subsequent exposures, thus gradually increasing drug resistance. Together, large target sizes for mutations and their Top1-guided generation lead to their gradual and rapid accumulation, synergistically accelerating the development of resistance.

## 1. Introduction

Resistance to chemotherapy is a leading cause of treatment failure. In line with prior knowledge of the development of antibiotic-resistant bacteria [1,2,3,4,5,6], the major concept in the cancer field is that resistance mutations happen randomly prior to treatment and are positively selected during the treatment. This notion was directly supported by experiments with cell barcoding that studied the selection of resistant clones, followed by the barcode analysis [7,8,9,10,11]. In these experiments, a fraction of clones selected in parallel independent treatments had the same barcodes, strongly suggesting that these clones originated from the same parental cells, which carried resistant mutations prior to the beginning of drug treatment [12,13,14]. However, other selected resistant clones that carried different barcodes were not further investigated in these studies and may result from mutations that were generated in response to drug treatment. Overall, in the selection of drug-resistant mutations in culture, cells are adapted to low drug concentrations, and then via multiple passages with dose escalation, resistant mutants are selected [6,15,16,17,18,19,20,21,22]. The process usually takes several months and provides resistance to higher than initial drug concentrations but not full drug resistance. Importantly, the selection of drug-resistant mutants in dose escalation experiments could be achieved from cloned genetically identical cell populations, suggesting that adaptation must involve the generation of mutations de novo in the process of drug treatment.

Exploration of forces driving drug resistance under dose escalation may help to uncover novel mechanisms of the development of resistance in the clinical setting [16,17,23,24,25,26,27,28,29,30,31,32]. Indeed, multiple administrations of drug doses in the clinic appear to mimic the in vitro scheme of drug resistance selection. Understanding the mechanisms of the development of drug resistance can also provide novel insights to guide the design of drug combinations and treatment strategies.

Here, we investigated how these adaptive mutations may emerge in colon cancer cells with the example of resistance to a widely used anti-cancer drug, irinotecan. Irinotecan is a pro-drug, which is converted into the active drug SN-38, which binds to Top1 [33,34,35,36]. The binding, in turn, allows Top1 to make DNA breaks but prevents re-ligation [34,37]. Top1-mediated single-strand breaks may facilitate double-strand DNA breaks that, if unrepaired, cause cancer cell death [33]. Top1 works both during transcription and replication [34]. A number of mechanisms of resistance to irinotecan have been described, including mutations in Top1 and the upregulation of multidrug-resistance pumps and their associated enzyme systems [36,38,39,40,41]. It is difficult, however, to understand why a dose escalation scheme could be important for the development of drug resistance based on these mechanisms. Here, we evaluated the efficiency of the development of irinotecan resistance and uncovered a novel resistance mechanism based on the active generation of a large number of mutations in Top1-dependent DNA breaking sites that reduce the chances of double-strand breaks upon consequent irinotecan exposures in the process of dose escalation.

## 2. Results

### 2.1. Experimental Design with Multiple Irinotecan (SN-38) Treatments

Since irinotecan is widely used against colon cancer, we investigated its effects on the colon cancer cell line HCT116. Since, unlike in the organism, in cell culture, irinotecan is activated very ineffectively, in all experiments below, we used an already activated derivative of irinotecan, i.e., SN-38. In order to achieve genetic uniformity in the population, we cloned the cells and isolated several independent clones. Genetic uniformity within each clone indicates that any drug resistance mutation selected in our experiments occurs either in the process of colony growth from a single cell prior to the drug treatment or is actively generated in the process of drug treatment. Sensitivity to SN-38 differed dramatically between the clones, with the minimal toxic concentration ranging between 1 nM and 80 nM (Appendix A). We chose two clones, SCC1 (single cell clone-1) and SCC7 (single cell clone-7), with high sensitivity (IC_50_ values of 1 nM and 2 nM, respectively) for further experiments.

To understand the development of drug resistance, SCC7 cells were exposed to 4 nM SN-38, which led to the death of a significant fraction of the population and the cell cycle arrest of the rest of the population. Cells remained in the arrested state without divisions for 14 days and then resumed growth. The arrest was associated with a senescence-like phenotype (highly enlarged and vacuolized cells). The second treatment with 4 nM SN-38 led to a shorter period of cell cycle arrest. Such a treatment cycle was conducted five times in total. At the fifth cycle, practically no cell death or growth inhibition was seen, indicating that cells became fully adapted to this concentration of SN-38 (Figure 1a,b).

Furthermore, we tested if a similar pattern occurs when cells develop adaptations to high doses (40 nM) of SN-38. The fraction of cells that survived 40 nM stayed in a senescence-like state for more than three months, after which cells resumed propagation and filled the plate. Upon subsequent exposure of the recovered population to 40 nM SN-38, the period of the growth arrest was only about one month. The process was repeated three more times, and each time, the fraction of dying cells was lower and the time period of growth arrest was shorter compared to the previous round of selection. Following the fourth exposure, cells spent approximately one week in the arrested state (Figure 1a,b). Altogether, these findings indicate similarities in adaptation to low and high doses of SN-38.

### 2.2. Most of the Survived Cells Recover from the Senescence-Like Growth Arrest

A large fraction of cells in the population could adapt to the treatment and resume growth after the cell cycle arrest. Alternatively, a small fraction of cells that could be originally resistant to the drug continued to propagate, which became detectable only when they began outgrowing the rest of the arrested population. To distinguish between these possibilities, SCC7 cells were infected with the cell cycle reporter virus [42] and exposed to 4 nM SN-38. Survived treated cells stopped dividing, acquired a senescence-like phenotype (enlarged, vacuolized cells), and according to the reporter, underwent G1 growth arrest (Figure 1c,d). Cells remained in G1 for four days, after which a majority of cells exited G1 and entered the cell cycle (Figure 1c,d). Entering the cycle following the senescence-like arrest was surprisingly slow, and only by day 8 had almost 100% of cells reached G2. There were no localized shifts of cells to G2, indicating the lack of clonal expansion. This observation indicates that cells underwent true adaptation to SN-38 rather than reflecting the expansion of a small fraction of initially resistant cells.

### 2.3. A Large Fraction of Cells Become Adapted to SN-38

To quantitatively assess the process of adaptation, 10 million SCC1 cells were individually barcoded using the Cellecta 50 M barcodes lentiviral library [43]. When cells filled the plate, half were collected for the barcode analysis, and half were used for further dose-escalation experiments with sequential passaging over 2, 4, 6, 8, and 15 nM of SN-38. With each passage, we observed a reduction in the population of dying cells and a reduction in the period of growth arrest (Figure 1e). We collected cells for the analysis of barcodes at different stages of the experiment, as described in Section 4.

A comparison of barcodes that were detected in the control population and the population treated with 2 nM showed that about 4 × 10^−3^ of the original clones survived the selection (Figure 1e). The analysis of barcodes in the population of cells after the 15 nM SN-38 selection demonstrated that 2 × 10^−3^ of the original clones survived the entire series of selections, which is only two times lower than the number of clones that survived the first round of selection at 2 nM. These findings indicate that (a) the probability of the survival of clones is high compared to the usual probability of spontaneous or even mutagen-induced mutations (according to classical studies, usually not higher than 10^−4^ [3,4,15,17,18,19,20,21,44,45]) and (b) almost 50% of clones that survive 2 nM selection survived the entire series of selections, suggesting that if cells are able to survive the initial treatment, they can also survive the dose escalation treatment.

To test if the dose escalation process is critical for the development of the resistant forms, barcoded SCC1 cells were exposed directly to 15 nM SN-38. Analysis of barcodes indicated that only 2 × 10^−5^ of clones survived (Figure 1e), which is 100× lower than the survival rate of 15 nM SN-38 in the dose escalation experiment. These data indicate that the dose escalation protocol is important for the effective development of resistant variants.

### 2.4. Changes in Transcriptome May Not Be Involved in Resistance Development

To explore the mechanisms of the adaptation, the population that survived multiple rounds of treatment with 40 nM SN-38 was cloned again. Barcodes from several clones were isolated and sequenced. Three clones with different barcodes (mutant single clone-MSC1, MSC2, and MSC3) were chosen for further analysis. Since the barcoding of cells was carried out prior to the entire series of SN-38 treatments, the fact that these clones carry different barcodes indicated that they did not split from the same clone somewhere in the middle of the SN-38 treatment. In other words, they represent the progeny of cells that underwent the entire series of treatments independently of each other. We would like to reiterate that the original barcoded population was genetically homogenous because of the initial cloning. Notably, the selected clones had a growth rate similar to the parental SCC1 clone (Appendix A).

To uncover potential mechanisms related to changes in gene expression, we compared transcriptomes of the parental SCC1 clone and the individual mutant isolates MSC1, MSC2, and MSC3, both untreated or exposed to 10 nM SN-38. The experiment was performed in biological duplicates (*n* = 2) for each sample. In naïve conditions, a number of differentially expressed genes were observed in the surviving clones (Appendix A). Importantly, we did not observe changes in either MDR1 or other ABC transporters involved in drug efflux, Top1, or DNA repair genes, suggesting that the mechanism of resistance in these clones may not be related to expected changes in the transcriptome. Furthermore, datasets were analyzed for the enrichment of genes belonging to known pathways using GSEA. We detected a number of pathways that were downregulated in the resistant mutants, such as epithelial to mesenchymal transition or inflammatory response pathway. These pathways are generally protective, and therefore their downregulation cannot explain the resistance of the selected mutants. Overall, these data clearly suggest that transcriptome changes are unlikely to be involved in resistance in the dose-escalation setting (Appendix A). 

### 2.5. DSBs Significantly Contribute to SN-38-Induced Cell Death

To further explore potential mechanisms of resistance, we performed a pooled shRNA screen to identify genes important for the survival of SN-38 treatment. SCC1 cells were infected with the focused lentiviral shRNA library Decipher Module-1 [46,47,48,49,50] that targets signaling pathways. This library covers about 20% of human genes. Cells were treated with 10 nM SN-38 for 24 h. Cells that survived the treatment after 5 days were collected, and the barcodes were isolated, sequenced and analyzed. We used the same population of infected cells but without SN-38 treatment as a control. Among the genes for which depletion showed sensitizing effects (Appendix A) was a group of genes that plays a role in DNA double-strand break repair, predominantly representing the homologous recombination (HR) and nonhomologous end joining (NHEJ) DNA repair pathways, including POLE, POLE3, POLE4, KAT5, RAD51C, RAD54L, RAD1, RAD9A, H2AFX, LIG4, and PARP2 (Appendix A). We also observed a number of genes involved in translesion DNA synthesis (Appendix A). The overall list of hits on the screen is shown in Appendix A. The list of pathways involved in the sensitivity according to the GSEA analysis of 1320 hits is shown in Appendix A (Appendix A), where the Hallmark DNA repair pathway, oxidative phosphorylation, Myc, and reactive oxygen species are in the 10 most highly enriched pathways. These data reinforce the understanding that (a) the generation of double-strand DNA breaks is critical for cell death caused by SN-38, and (b) the HR and, to some extent, NHEJ repair pathways play an important role in SN-38 survival in naïve cells (though these pathways are not significantly upregulated in the resistant clones (Appendix A)).

### 2.6. Development of SN-38 Resistance Associates with Emergence of Multiple Non-Random Mutations

To study mutations that emerged in the survived clones, we performed whole genome sequencing. Genomes of the survived clones MSC1, MSC2, and MSC3 were compared with the genome of the original population of SCC1, and each of them was compared with the reference genome (GRCh38/hg38) in the UCSC [51] database. We observed that the parental SCC1 genome had hundreds of thousands of SNPs and InDels compared to the reference genome.

These mutations may reflect the fact that HCT116 cells were isolated from a different individual than a group of individuals whose sequences compose the reference genome. Alternatively, these mutations could arise in the process of cancer development and/or further culturing of HCT116 cells in laboratory conditions. The overall analysis of mutations in SCC1 indicated that 93% of them do not correspond to known SNPs in the human population, suggesting that the vast majority of the mutations simply reflect either the cancer nature of these cells or genetic instability upon culturing. Accordingly, they will be further called “cancer alleles”. Notably, HCT116 cells carry a mutation in the MLH1 gene that leads to microsatellite instability [52,53,54], which may significantly contribute to the generation of these cancer alleles; see below. When genomes of the SN-38-resistant isolates MSC1, MSC2, and MSC3 were compared with the genome of SCC1, we identified hundreds of thousands of InDels and SNPs that arose in the process of adaptation to SN-38.

Strikingly, a very large fraction of these mutations were common between the independently isolated clones (Figure 2a,b). If compared by pairs, i.e., MSC1/MSC2, MSC1/MSC3, and MSC2/MSC3, in each pair, between 17% and 45% of mutations were common, and between 7% and 15% were common between all three independent isolates (46,099 mutations, of which 28.4% were InDels, and 71.6% were point mutations (Appendix A)). Even considering that SN-38 may have high mutagenic activity and triggers protective mutations with a rate as high as 10^−4^ per nucleotide, the probability of overlap of a mutation in three independent clones will be 10^−12^ (i.e., much less than one triple mutation per 3 × 10^9^ bp genome), which is many orders of magnitude lower than seen in the experiment. Thus, the overlapping (common between three clones) mutations clearly point to a non-random mutation mechanism. The generation of these mutations seems to be guided by a mechanism, an understanding of which may clarify the adaptation pathway.

Analysis of the ENCODE (promotors and enhancer datasets [55]), GEO (GSE57628, mapping of Top1 sites in human HCT116 cells) [56] and UCSC [51] database datasets uncovered that the vast majority of the mutations are present in heterochromatin (high content of histone H3K9me3) in the non-coding regions (Figure 2c). There was a small fraction of mutations present in promotors/enhancers (<1%) and the coding regions (2%), with most of them in the exons (Figure 2b,c). Importantly, the genes that have mutations in their coding and regulatory regions did not belong to known pathways associated with either cell survival or DNA repair and therefore are unlikely to be involved in the adaptation process (Appendix A; also see the mutation landscape section in the Appendix A for a description of an exceptional case). To avoid the dilution of the focus of this paper, we present a more detailed analysis of mutations in the Appendix A mutation mapping to regions in the genome and its correlations to distinct signatures such as the nature of repetitive elements (Appendix A), the identification of pericentromeric and heterochromatin regions based on methylation signatures of H3k9me3 and H3k27me3 (Appendix A), and a genome-wide density plot for triple mutations (Appendix A)). Altogether, these data suggest that adaptation to SN-38 was not associated with either mutations in functional genes or changes in the expression of these genes.

### 2.7. Mutations Result from Repair of Top1-Generated DSBs

To understand how hundreds of thousands of mutations in repeats and untranslated regions (see Appendix A) could be involved in adaptation to SN-38, we proposed that they could result from DNA breaks generated by Top1. Indeed, the mutation sites strongly co-localized with sites of DNA cleavage by Top1 [56] (19.8% of cases). By applying a hypergeometric distribution algorithm (see Section 4), we established that such co-localization is highly statistically significant (*p* << e^−198^). This percent of overlap is probably a strong underestimation since experimental conditions in the two studies were different, i.e., long-term generation of mutations vs. short-term Top1 inhibitor treatment. Furthermore, when we overlaid the distribution of triple mutations along the chromosomes over the distribution of Top1 cleavage sites, we observed a strong overlap of peaks (Figure 2d). Therefore, mutations take place at Top1 cleavage sites.

There are several lines of evidence suggesting that most of the mutations were generated either by the HR or NHEJ repair of DSBs. A total of 16% of the mutations in the isolates were clustered (7510 out of 46,099 triple mutations sample), where 2–7 mutations were present within regions of up to 100 bp (we chose this length of DNA for the definition of clustered mutations; see Section 4 (Figure 3a and Appendix A)). Since random positioning predicts that mutations should be, on average, separated by about 10,000 bp (about 300,000 mutations per clone distributed over 3 billion base pairs of the total genome), such clustering demonstrated their non-random appearance and suggested the mechanisms of their generation. In all these cases, these were loss of heterozygosity (LOH, herein referred to as reversion from a mutated allele to a reference allele) mutations. These LOH mutations did not result from deletions of one of the alleles (since the number of reads corresponding to these regions was similar to the average number of reads along the genome) but by copying an allele from one chromosome to another, including copying the entire mutation cluster (Figure 3b). Such copying could result only from the HR repair of DSBs. LOH occurred in 78% of common mutations (36,228 mutations out of 46,099), and a similar fraction of LOH was found with the overall set of mutations in resistant clones (Appendix A), suggesting that the majority of mutations were generated by the HR repair system.

In the resistant clones, 28% of de novo mutations that were not LOH resulted from NHEJ (9871 de novo common mutations, and a similar fraction of NHEJ was found in the overall set of mutations). They resulted from NHEJ because, in the case of insertions, these mutations have a very specific signature of duplication of a neighboring region. This duplication results from pairing broken ends at the terminal nucleotides and filling the gaps on both strands via translesion DNA synthesis (Figure 3b). This structure allows precise identification of the site of the DNA break, i.e., at the site of pairing between the duplication regions (Figure 3b) (see also Appendix A). Overall, de novo InDels can be used as hallmarks of DNA breaks that were repaired via NHEJ, while LOH mutations can be used as hallmarks of DNA breaks repaired via HR.

Very importantly, a high fraction of the overlapping mutations between independent resistant clones suggests that the SN-38-inhibited Top1 generates DNA breaks at specific sites in the chromatin (possibly specific Top1 binding or activation sites), which further leads to the generation of mutations upon the HR or NHEJ repair of DSBs.

### 2.8. Mechanism of Adaptation to SN-38

The following considerations provide a framework for understanding the mechanism of adaptation. At each genome location, the parental SCC1 population could have alleles either with no mutations compared to the reference genome (0/0), with mutations in one allele (0/1) (heterozygosity) or both alleles (1/1) (Figure 4a,b). Sites where two alleles in SCC1 had different mutations compared to the reference genome (1/2) were extremely rare. Accordingly, mutations that appear in MSC clones compared to parental SCC1 could be mutations de novo generated by NHEJ (e.g., 0/0→0/1) or loss of heterozygosity generated by HR (0/1→0/0; or 0/1→1/1) (Figure 4c,d).

In sites with LOH repaired by HR, a heterozygous allele could revert to either the reference genome allele (0/1 to 0/0) or to the “cancer” allele seen in the parental clone SCC1 (0/1 to 1/1). A surprising key observation that led to understanding the mechanism of adaptation was that the frequency of 0/1 to 0/0 shifts was 5.44 times higher than 0/1 to 1/1 shifts (Figure 5a). A 0/1 to 0/0 shift means that the allele from the reference genome was copied to the DSB at the “cancer” allele (an allele in SCC1 parental cells that differs from the reference genome), which ultimately means that the probability of double-strand DNA breaks in “cancer” alleles is 5.44 times higher compared to the reference genome allele. The probability of breaks in the “cancer” allele was even higher in the pericentromeric regions, where the ratio of breaks in the reference genome allele to the “cancer” allele was 1/20 (Figure 5b). In the chromosome arms, this ratio was 1/3.4. Therefore, surprisingly, alleles that acquired mutations in the course of cancer development were significantly more prone to Top1-induced double-strand breaks than normal human genome alleles (Figure 5b).

This unexpected finding provides the mechanism of gradual adaptation to SN-38. Initial exposures to SN-38 generate reversion of a number of “cancer” alleles to the reference genome alleles, which are more resistant to Top1-induced DSBs, which in turn creates a protective mechanism against subsequent exposures. In other words, with each exposure, there are fewer and fewer potential Top1-cleavage sites, which leads to stronger and stronger adaptation. This novel mechanism of adaptation to SN-38 does not involve the expression of any protective proteins or mutations, but rather involves a high number of changes in the DNA structure that make it less prone to Top1-induced breaks.

An important consequence of this mechanism is that the cells become adapted specifically to SN-38 and do not develop resistance to other genotoxic agents. Indeed, the SN-38-resistant clones remained highly sensitive to the intercalating agent inhibitor of Top2, i.e., doxorubicin (Appendix A).

### 2.9. Adaptation Associates with Reduced Ability of SN-38 to Trigger DNA Breaks

This mechanism predicts that in the process of adaptation, following multiple exposures to the same concentration of SN-38, cells should experience fewer DSBs, while the rate of repair of DSBs remains the same. To test this prediction, we took SCC7 cells that had not been drug-exposed and that underwent five cycles of exposure to 4 nM of SN-38. Both populations were subjected to 4 nM of SN-38 for 24 h, and the number of γH2AX foci was assessed by immunofluorescence. While SN-38 exposure of naïve cells caused a dramatic increase in the number of foci, exposure of cells that were preadapted by five cycles of SN-38 treatment barely caused foci formation (Figure 6a,b). On the other hand, the rate of DSB repair (recovery of γH2AX foci) was not accelerated (Figure 6c). Importantly, lower foci formation correlated with the lack of cell death and growth arrest. Similarly, there was a lower overall number of DNA breaks in adapted cells, as judged by the comet assay (Figure 7a–e). A similar experiment was carried out with parental SCC1 and SN-38-adapted MSC1, MSC2, and MSC3 clones, but instead of 24 h, exposure to the drug (40 nM) lasted for durations of 3, 6 and 12 h. Figure 8a shows that the overall phosphorylation of γH2AX level gradually increased in the course of the experiment, as shown with an immunoblot. As with the SCC7 experiment, the drug-adapted mutants showed much fewer DSBs than the parental clone at 24 h of exposure (Figure 6a,b). Similar effects were seen upon quantification of the number of γH2AX foci in SCC1 and the mutants. Representative images for 24-h treatment are shown (Figure 8b,c) considering the highest difference in γH2AX levels on immunoblots (Figure 8a). A similar experiment was performed with doxorubicin to quantify the γH2AX foci after 24 h of treatment. We observed foci generation in both SCC1 and SN-38-resistant mutants (Figure 8b,d). In order to check the foci generation at the lowest time point, we chose a short 30 min treatment for SCC1 and mutants and quantified the γH2AX. At this time point, we also observed significant differences in the foci formed in SCC1 and mutants (Appendix A). Therefore, indeed, the reversion of “cancer” alleles to the reference genome alleles appears to be associated with fewer DSBs by SN-38.

Another prediction from the suggested mechanism of resistance was that in the MSC mutants, the ability of Top1 to interact with DNA is reduced compared to the parental SCC1 clone. Therefore, we sought to compare the amount of Top1 covalently bound to the chromatin following SN-38 treatment in SCC1 and MSC1 clones using the DNA-Top1 adduct capturing RADAR assay. Indeed, in the presence of SN-38, in resistant mutant MSC1, the amounts of Top1-DNA adducts were significantly lower than in SCC1, further indicating that this drug adaptation is associated with the loss of Top1 cleavage sites (Figure 9a,b).

## 3. Discussion

Here, we addressed why the approach toward the selection of drug-resistant mutants in cancer cells requires multiple exposures to drugs and dose escalation. Such a selection scheme suggests that (a) either cells are somehow adapted to the low concentrations of drugs (e.g., via epigenetics mechanisms), and this adaptation guides the further selection of the resistant mutant forms; or alternatively (b), the development of drug resistance involves acquiring a large number of mutations, each of which provides a fraction of the resistance, but gradually, they accumulate and are selected in the process of drug escalation. At least with an inhibitor of Top1, irinotecan, we show that the second possibility is correct. It appeared that a very high number of mutations was generated in the process of the selection of irinotecan-resistant mutants in the dose escalation experiment. Surprisingly, the absolute majority of them were in the non-coding and silenced regions of the genome, suggesting that these mutations do not affect the expression or function of specific genes involved in irinotecan resistance. Accordingly, this mechanism of adaptation is fundamentally different from previously known mechanisms related to changes in drug targets, drug metabolism, or transport.

During the adaptation process, we observed that surviving cells acquired a senescence-like phenotype and stayed in such a state without divisions for weeks and even months until the population resumed divisions. Since senescence-like or other types of dormant states can provide drug resistance [57,58,59,60], it is possible that the acquisition of this phenotype plays a role in the ultimate survival of drug treatment.

The key to understanding the nature of resistance was the observation that these mutations result from the repaired DSBs via the HR and NHEJ pathways. Though cuts by Top1 generate single-strand breaks (SSB), these breaks can develop into DSBs [34]. In HR-dependent DSB repair, we observed loss of heterozygosity associated with the copying of intact alleles to the allele with the DSB, which allowed the precise identification of the allele with the DSB. Strikingly, more than 80% of DSBs took place in the alleles with mutations associated with the cancer nature of the parental cells, and accordingly, the HR repair led to the restoration of the original “normal” alleles. As an example of such sequences, we found a large fraction of breaks in the polyC sequences of 30–40 bp, which were present in the parental cancer cells (see Appendix A). As a result of HR repair, these alleles were changed to alleles with interrupted polyC regions, which were present in the reference human genome. Accordingly, at these sites, DSBs appear to require extended polyC, and thus, an allele that has an interruption of the polyC tract must have a lower probability of breaks. Therefore, the reversion of extended polyC to the interrupted tract of polyC protects this site from further breaks and thus contributes to the overall development of resistance to SN-38 (Appendix A). This observation ultimately means that upon the first exposure to the drug, a fraction of sites with a high probability of Top1-induced breaks will be reverted to sites with a low probability of breaks. Therefore, with each cycle of exposure to SN-38, fewer and fewer sites with a high probability of DSBs will remain in the genome, which ultimately must increase the chance of survival. Indeed, we demonstrated that the number of DSBs is reduced following cycles of exposure to SN-38, and this effect was associated with a lower probability of breaks rather than with more efficient DNA repair. Therefore, the development of resistance to SN-38 involves acquiring a large number of mutations, each of which provides a fraction of the overall resistance, which gradually accumulate and are selected in the process of drug escalation.

This adaptation mechanism is associated with a gradual reduction in the number of DNA breaks by Top1, suggesting a lower efficiency of Top1-dependent relaxation of DNA supercoils in selected clones. Possibly, the number of mutations that provide adaptation to SN-38 may be limited by the necessity to carry out the DNA relaxation activity. Alternatively, in these clones, Top2 can take over essential DNA relaxation (Figure 9 and Appendix A).

Unexpectedly, the majority of the Top1 cleavage sites were in the satellite regions, especially in short repeats. This feature may explain a well-known fact that microsatellite instability in colon cancer is associated with a better response to irinotecan therapy [61,62,63]. Indeed, we suggest that microsatellite instability may generate additional Top1 sites, which makes these cells more sensitive to irinotecan. Notably, the HCT116 cells that we used in the experiments have microsatellite instability [54,63,64].

Furthermore, our findings may explain a very unusual feature of irinotecan. Unlike almost any anti-cancer drug, irinotecan is not effective against stage I or II cancers and becomes quite effective in advanced and metastatic cancers [36,65,66,67,68]. This puzzling feature is probably related to genetic instability in cancer, which generates additional Top1 cleavage sites during cancer evolution towards advanced stages, thus making cells more sensitive to irinotecan.

This work also illuminates novel aspects of the function of Top1. Though it was reported that Top1 associates with active RNA polymerase to relieve DNA supercoils generated in the process of transcription [56,69,70], our data suggest that Top1 can also function in a transcription-independent relief of supercoils since a majority of mutations was seen in heterochromatin (H3K9me3 and H3K27me3 patterns; see Appendix A). The fact that there was a very large fraction of DSBs in common among the three independent SN-38-resistant isolates indicates that there are preferable sites of breakage. This idea is reinforced by the finding that the mutation sites highly overlapped with sites of DNA cleavage by Top1 [56]. Possibly, these mutation sites are preferable sites of binding of Top1 or some Top1 activating factors to DNA, suggesting that Top1 or its activators have a sequence-binding preference. Alternatively, Top1 may bind anywhere on the DNA and, when moving together with RNApol and possibly DNApol, stalls at these regions to increase the probability of cuts. Another attractive possibility is that these repeat regions are preferable sites where SSBs are converted to DSBs.

A prior published study indicated that Top1 binds to DNA sequences non-specifically. Experiments with topotecan, however, demonstrated that cleavage takes place at relatively specific sites [56]. Surprisingly, though Top1 was implicated in transcription [56], Top1-generated cleavage took place both in transcriptionally active and inactive regions, including centromeres (Figure 3d). In line with this finding, the published Top1 cleavage sites overlapped with irinotecan-generated mutation sites mostly in silenced regions (Figure 3d). Accordingly, it appears that besides transcription, Top1 may also serve in other processes, e.g., replication or repair.

This novel drug resistance mechanism may have interesting implications for understanding evolutionary processes. Indeed, it is possible that DSBs generated by Top1 take place predominantly at the sites of mutations that deviate from “normal” genomes. Accordingly, these DSBs can be repaired by the HR pathway, which will lead to the restoration of normal genome homozygosity (ref abstract figure). In other words, a stabilizing evolutionary selection may take place even in the absence of selection pressure, simply as a result of the Top1 and HR repair function. Accordingly, the overall diversity of SNPs and InDels in the plurality of normal genomes has limitations that are shaped by the function of the Top1 and HR repair systems.

## 4. Materials and Methods

### 4.1. Cell Culture and Reagents

Cell lines were obtained from the ATCC. HCT116 cells (ATCC Cat# CCL-247, RRID:CVCL_0291) were cultured in McCoy’s 5A medium supplemented with 10% FBS, 4 mM L-glutamine (Cat#03-020-1B, BI-Biologicals, Ingelheim, Germany), 2 mM L-alanyl-L-glutamine (Cat#03-022-1B, BI-Biologicals), and 1% penstrep (Cat#03-031-1B, BI-Biologicals) and were grown in a humidified incubator at 37 °C and 5% CO_2_ [71]. SN-38 was purchased from Sigma-Aldrich (St. Louis, MO, USA).

### 4.2. Single Cell Line Cloning

Single-cell cloning was performed with the limiting dilution method. Individual clones were isolated using cloning discs, (Merck, Darmstadt, Germany, Ca#Z374431), then grown and stored (10% DMSO in FBS) as stocks in liquid nitrogen for further use.

### 4.3. Cell Cytotoxicity Assay

Cells were plated at 30% confluency and treated with the drug at the specified concentrations according to the experimental plan. Following treatment for a specified period of time, the drug was removed, and cells were washed with 1x-PBS, followed by fixation with 1.2% formaldehyde for 10 min. After washing, cells were stained with DAPI (1:5000 dilution) for 5 min and washed 4 times with 1x-PBST. Imaging was performed using the Hermes Wiscon Imaging System (IDEA Bio-Medical Ltd., Rehovot, Israel, WiScan Hermes High-Content Imaging System, RRID:SCR_021786), and image analysis was performed using an inbuilt software package system (Athena Wisoft, Ver. 1.0.10) called the “count cell algorithm”. DAPI-positive fixed cells were counted and compared to the untreated control to quantify the drug response as described [72].

### 4.4. Virus Preparation for the Barcoding Library and shRNA Screens

Lentiviral libraries for barcoding and shRNA screens were prepared according to the manual. Briefly, cells were passaged and grown at 80–90% confluency. For transfection, reagents, plasmids, and Lipofectamine 3000 were mixed in opti-MEM (Thermo Scientific, Waltham, MA, USA) supplemented with 4 mM glutamine and co-incubated overnight. The next day, the media was changed with OptiMEM supplemented with 5% FBS and 4 mM glutamine and kept for 24 h. Viruses were harvested using a 0.45 μm filter and kept at −80 °C until further use. Upon infection with corresponding libraries, we chose an MOI of about 20% so that, on average, each cell received only one viral particle. After infection, cells carrying lentiviruses were selected with puromycin and further divided into groups for drug treatment in culture [73].

### 4.5. Cell Barcoding

The cloned cell population was barcoded with the 50 million (50 M) library according to manufacturer protocols (Cellecta CloneTracker 50M Lentiviral Barcode Library, RRID:SCR_021827 [43]). Briefly, cells were infected with the barcoding lentiviruses, selected with puromycin, and further divided into groups for drug treatment. After the treatments, cells were allowed to recover, genomic DNA was purified, and barcodes were isolated by nested PCR. All samples were multiplexed for sequencing. The detailed procedure of PCR and primer details for the 50 M library are available in the Appendix A).

### 4.6. Pooled shRNA Genetic Screen

For the shRNA screen, we used the human decipher module 1 library (RRID:Addgene_28289) [46,47,48]. Cells were infected with this pooled shRNA library with a low MOI. Cells were treated with SN-38 for 24 h, recovered for 4 days, and collected for DNA isolation and further processing. Barcodes were isolated by nested PCR, then sequenced and analyzed according to the manufacturer’s protocol.

### 4.7. Genomic DNA Extraction and Amplification of Library Barcodes

The isolation of genomic DNA from cultured cells was performed by a Wizard genomic DNA isolation kit (Promega, Madison, WI, USA). Amplification of the barcodes was carried out by nested PCR. The detailed procedure of PCR and primer details for theshRNA screen are available in the Appendix A. Briefly, the first PCR (PCR 1) was performed using Titanium Taq DNA Polymerase (# 639209, Takara Bio, San Jose, CA, USA). Separation of the PCR products from primers and gel purification was carried out with a QIAquick PCR & Gel Cleanup Kit (Qiagen, Hilden, Germany). The second PCR (PCR 2) was carried out using nested primers, which were either generic or had unique sample barcodes. PCR 2 was performed using the Phusion High-Fidelity PCR Master Mix (Thermo Scientific, Waltham, MA, USA). Samples were multiplexed by adding an additional sample barcode during the second round of PCR. Samples were normalized individually, then pooled together, and purification of the PCR products was completed using AmpureXP magnetic beads (Beckman Coulter, Brea, CA, USA) following manufacturer protocols. Next, we sequenced the barcodes using Ion Torrent.

### 4.8. Analysis of the Barcode Data

We used a combination of custom-tailored applications to analyze sequencing reads along with the R packages. Data were first checked for quality of reads through FastQC (v0.11.7, RRID:SCR_014583). Then, using a barcode splitter (v0.18.6, barcode splitter, RRID:SCR_021825), reads were demultiplexed based on sample barcodes (1 error as mismatch or deletion was allowed for sample barcodes while demultiplexing). The obtained FASTQ files were used to count the library barcodes using Python-based applications that were custom-made for this purpose. Quantification of the unique barcodes that were enriched or lost after treatment was carried out via a Python-based script (software version 3.10.0, RRID:SCR_008394). For data cleaning and visualization, the tidyverse-v1.0.0 (RRID:SCR_019186) and ggplot2-v3.3.3 (RRID:SCR_014601) [74] R packages were utilized.

### 4.9. Transcriptome Analysis

RNA was extracted from cells using the RNeasy Mini kit (Cat#74104, Qiagen). A library preparation strategy (BGISEQ-500, RRID:SCR_017979) was adopted and performed by BGI, China. Briefly, mRNA molecules were purified from total RNA using oligo(dT) attached magnetic beads and fragmented into small pieces using a fragmentation reagent after reacting for a certain period at the proper temperature. First-strand cDNA was generated using random hexamer-primed reverse transcription, followed by a second-strand cDNA synthesis. The synthesized cDNA was subjected to end repair and then was 3′ adenylated. Adapters were ligated to the ends of these 3′ adenylated cDNA fragments. This process amplified the cDNA fragments with adapters from the previous step. PCR products were purified with Ampure XP Beads (AGENCOURT) and dissolved in EB solution. The library was validated on an Agilent Technologies 2100 bioanalyzer (2100 Bioanalyzer Instrument, RRID:SCR_018043,). The double-stranded PCR products were heat-denatured and circularized by the splint oligo sequence. The single-strand circle DNA (scir-DNA) were formatted as the final library. The library was amplified with phi29 to make DNA nanoballs (DNB), which had more than 300 copies of one molecule. The DNBs were loaded into the patterned nanoarray, and 50 single-end (100 pair-end) base reads were generated in the way of sequencing by synthesis. An in-house pipeline was developed to analyze the data, where reads were first trimmed and clipped for quality control in trim_galore (v0.5.0, RRID:SCR_011847), then checked for each sample using FastQC (v0.11.7, RRID:SCR_014583). Data were aligned by Hisat2 (v2.1.0, RRID:SCR_015530) using hg38 and GRch38.97. High-quality reads were then imported into samtools (v1.9 using htslib 1.9, RRID:SCR_002105) for conversion into SAM files and later to BAM files. Gene-count summaries were generated with featureCounts (v1.6.3, RRID:SCR_012919), as follows. A numeric matrix of raw read counts was generated, with genes in rows and samples in columns, and used for differential gene expression analysis with the Bioconductor RRID:SCR_006442 (edgeR v3.32.1, RRID:SCR_012802 [75], Limma v3.46.0, and RRID:SCR_010943) packages to calculate the differential expression of genes. For normalization, the “voom” function was used, followed by the eBayes and decideTests functions to compute the differential expression of genes.

### 4.10. Human Whole-Genome (HWG) Sequencing and Analysis

Isolated DNA were sent for 30x whole-genome sequencing as a service by the commercial provider Dante Labs, Italy. After obtaining reads, data were first checked for the quality of the reads through FastQC (v0.11.7, RRID:SCR_014583). Then, using the barcode-splitter (v0.18.6), reads were demultiplexed based on sample barcodes (1 error -as mismatch or deletion was allowed for sample barcodes while demultiplexing). Reads were aligned to the reference genome hg38 with BWA-MEM (RRID:SCR_010910) [76] default settings. Aligned SAM files were converted to BAM files using samtools (ver1.9, RRID:SCR_002105). The obtained BAM files were used to call variants using the Genome Analysis Tool Kit (GATK v4.1.8.0, RRID:SCR_001876) [77] from Broad’s Institute; we called for combined genomic variants in the form of variant call factor (VCF) files as output. In the above analysis, we strictly followed the best practices guidelines of GATK [78]. All further analysis was done using suitable R packages (tidyverse-v1.0.0 (RRID:SCR_019186), ggplot2-v3.3.3 (RRID:SCR_014601), or karyoploteR v1.18.0 (karyoploteR (RRID:SCR_021824)) [79]. For the data, we also used the SnpEff (v3.11, RRID:SCR_005191) [80] program for the annotation of genes and bedtools v2.27.1(RRID:SCR_006646) [81] to match obtained variants to available bed files from the UCSC (UCSC Cancer Genomics Browser, RRID:SCR_011796) [51] and GEO (Gene Expression Omnibus (GEO), RRID:SCR_005012) datasets and other relevant databases. To extract data from metadata files and for comparative analysis, we used Microsoft Excel (RRID:SCR_016137). Raw data files and associated common mutation analysis in the form of VCF files are available at SRA (NCBI Sequence Read Archive (SRA), RRID:SCR_004891) under the accession number (PRJNA738674).

### 4.11. Assessment of Cluster Mutations in Resistant Mutants

For estimating the number of mutations present in close proximity (clusters), we took 100 bp small windows and calculated whether the mutations were present in close proximity in these small intervals. We used custom-written Python codes to first separate the whole genome into small bins of 100 bp. Furthermore, we defined the genomic locations of common mutation’s genomic locations and calculated how many of these mutations fell within the 100 bp window. The output was generated as a text file, where the genomic locations and the number of mutations in the 100 bp window were calculated.

### 4.12. Classification of Repeats and Computing Mutations in Low-Complexity Regions of Genome

For classifying the nature of repeats where these mutations accumulated, we used RepeatMasker v4.1.0 (RepeatMasker, RRID:SCR_012954, database Dfam_3.1 (RRID:SCR_021168) and rmblastn version 2.9.0) [82], which screens DNA sequences for known repeats and low-complexity DNA sequences. The reference sequence hg38 was used for this purpose. Common (triple) mutations (total-46099) were aligned to the reference database according to their occurrences based on genomic locations. The program we employed defines the nature of the sequence where these mutations have accumulated. The output of the program is a detailed annotation of repeats that are present in the query sequence, as well as a modified version of the query sequence in which all the annotated repeats are classified. Sequence comparisons were performed in a Unix environment using cross-match, an efficient implementation of the Smith–Waterman–Gotoh algorithm developed by Phil Green, or using WU-Blast, developed by Warren Gish [82]. The output file was a matrix of repeated elements found in the query sequence compared to the reference libraries, reported as percentages in tabulated form [83].

### 4.13. Assessment of Satellites and Simple Repeats

An in-depth analysis was carried out to identify the nature of satellites and their classification using RepeatMasker v4.1.0 (RepeatMasker, RRID:SCR_012954) [82] and its advanced options, wherein we investigated the types of satellites and simple repeats. Firstly, fasta sequences for the genomic locations for common mutations were fetched using the bedtools “getfasta” algorithm (BEDTools, RRID:SCR_006646) [81]. These were further analyzed by RepeatMasker v4.1.0 [82] to identify the repeated elements. Next, “fasta.out” was used to further investigate and quantify the classification of the satellite types. Subsequently, “fasta.tbl” was used to compute the overall percent representation of the repeated elements. Additionally, we used the tandem repeat finder (RRID:SCR_005659) [84] and the UCSC microsatellite track to investigate and compute the microsatellites and simple repeats.

### 4.14. Double-strand Break Quantification

The estimation of double-strand breaks occurring in the wild-type cell population and resistant lines once adapted to the drug was carried out using (a) a γH2AX assay [85] and (b) a comet assay [86].

### 4.15. γH2AX Assay

SCC7 parental cells and cells that underwent five cycles of drug exposure (T5_resistant) were exposed to 4 nM SN-38 for 24 h. Alternatively, SCC1 and MSC1 cells and MSC2 and MSC3 cells were exposed to 40 nM SN-38 for 30 min. Cells were washed with 1x-PBS and fixed with 0.2% formaldehyde. Permeabilization was carried out with 0.2% Triton X-100 in PBS for 10 min at room temperature. Then, cells were blocked with bovine serum albumin (BSA, 5% *w/v* in phosphate-buffered saline with 0.05% Tween-20 (PBST) for 1 h. Cells were then washed 3 times with PBST. Incubation with the primary antibody (Phospho-Histone H2A.X (Ser139) Antibody, Cell Signaling Technology Cat# 2577, RRID: AB_2118010) occurred overnight at 4 °C. Cells were then washed with 1x-PBST five times, followed by incubation with the secondary antibody (Goat Anti-Rabbit IgG H&L, Alexa Fluor 488, Abcam Cat# ab150077, RRID:AB_2630356) for 1 h. The antibody was removed, and cells were washed five times to remove non-specific antibody binding. Imaging was performed using a Hermes Wiscon Imaging System (IDEA Bio-Medical Ltd., WiScan Hermes High Content Imaging System, RRID:SCR_021786), and image analysis was performed using an inbuilt software package system (Athena Wisoft, Ver1.0.10). The software takes the maxima and minima for the foci intensity. For all the analysis, we kept constant maxima = 550 to select the foci and selected automatic background correction based on its untreated sample. Statistics were calculated with GraphPad Prism version 9.0.0 (GraphPad Prism, RRID:SCR_002798), California, USA. The significance of differences was determined using an unpaired Welch’s correction and two-tailed t-test (“ns” < 0.1234, * *p* < 0.0332, ** *p* < 0.0021, *** *p* < 0.0002, and **** *p* < 0.0001).

### 4.16. Single-Cell Gel Electrophoresis (SCGE, Alkaline)

To estimate the change in DNA strand breaks before and after treatment in normal cells and resistant cells, a single-cell gel electrophoresis (SCGE) or COMET assay was performed [87]. Cells were embedded in 1% agarose on a microscope slide and lysed with detergent and high salt.

Glass slides were precoated with 1% agarose. Cells were treated with SN-38 or left untreated to serve as controls. The preparation of the sample was carried out by scraping cells gently with 0.05% trypsin. Cells were then washed 3 times with ice-cold 1x-PBS, and roughly 0.1 × 10^6^ cells/mL were taken. A total of 50 uL of cell suspension and 50 uL of 1% low molten agarose kept at 37 °C were mixed together. Then, 75 uL was used to make bubbles on the slide and left to solidify at 4 °C until it formed a clear ring. Lysis was performed by placing the sample slides in lysis buffer (Nacl (2.5 M), EDTA-pH 10 (100 mM), Tris-Base pH 10 (10 mM), and Triton X100 (1% freshly added before use); buffer pH was maintained at 10 before adding Triton-X100) overnight at 4 °C. Unwinding was performed by rinsing the slides with fresh water to remove salts and detergents. Slides were immerged in unwinding buffer (NaOH (300 mM) and EDTA (1 mM); buffer pH 13) for 1 h at 4 °C. After unwinding, the samples were run in an electrophoresis alkali buffer ((NaOH (12 g/L) and EDTA (500 mM pH 8)). Slides were kept in an electrophoresis tank, and buffer was poured to cover the slides. Running was performed at 22 V (constant) and 400 mA (constant) for 40 min. Neutralization was performed by dipping the slides in Tris buffer (0.4 M, pH 7.5) for 5 min. Slides were then immersed in 70% ethanol for 15 min and air-dried for 30 min. Staining was performed by incubating cells in DAPI (1 - μg/μL, 1:5000 dilution). Cells were then washed with ice-cold water and air-dried in the dark. Imaging was performed using an Olympus IX81 Inverted Fluorescence Automated Live Cell Microscope (18 MP CMOS USB camera) with the associated built-in software package (Olympus IX 81 Inverted Fluorescence Automated Live Cell Microscope, RRID:SCR_020341). Quantification was performed using OpenComet v1.3.1 (OpenComet, RRID:SCR_021826) [88]. Statistics were calculated with GraphPad Prism version 9.0.0 (GraphPad Prism, RRID:SCR_002798), California, USA. The significance of differences was determined using unpaired Welch’s correction and two-tailed *t*-test (* *p* < 0.0332, ** *p* < 0.0021, *** *p* < 0.0002, and **** *p* < 0.0001).

### 4.17. Cell Cycle Reporter Assay

The reporter plasmid (pBOB-EF1-FastFUCCI-Puro, Addgene-plasmid #86849; http://n2t.net/addgene:86849; RRID:Addgene_86849, accessed on 12 July 2021) was made by Kevin Brindle and Duncan Jodrell [42]. Lentiviruses were produced in the lab as described in Section 4 above. Cells were infected with the viruses and selected with puromycin. Treatment was carried out with SN-38 (4 nM) for 24 h. After drug removal, recovery was recorded by live imaging on alternate days. Control cells refers to cells before treatment, and images were taken on consecutive days until the 9th day of recovery. Analysis was performed using the FIJI-ImageJ program using imaged channels.

### 4.18. DNA Top1 Adduct Capturing RADAR Assay

The HCT116 (SCC1 parental clone) and mutant SCC1 (MSC1 and MSC2) cell lines were seeded in 6-well plates in 2 mL of McCoy’s 5A medium supplemented with 10% FBS, 4 mM L-glutamine (Cat#03-020-1B, BI-Biologicals), 2 mM L-alanyl-L-glutamine (Cat#03-022-1B, BI-Biologicals), and 1% penstrep (Cat#03-031-1B, BI-Biologicals) at 25% confluence and were cultured overnight in a humidified incubator at 37 °C and 5% CO_2_. The next day, the medium was aspirated, and the cells were washed with 1x-PBS and treated with 40 nM irinotecan for 6 h. After treatment, the medium was removed, and cells were lysed on the plate by the addition of 3 mL of the lysis reagent RLT Plus (Lot 169010027, Qiagen, Germany). A total of 1 mL of lysate was transferred to a 2 mL Eppendorf tube, and 0.5 mL of 100% ethanol (1/2 volume) was added. The mix was incubated at −20 °C for 5 min and centrifuged for 15 min at 16,000 rcf. Another volume of lysate was frozen and stored at −80 °C. After centrifugation, the supernatant was aspirated, and the pellet containing nucleic acids in complex with proteins was washed twice in 1 mL of 75% ethanol by vortexing, followed by 10 min of centrifugation. Finally, the pellet was diluted with 20 μL of 8 mM NaOH and 20 μL of Tris-buffered saline buffer (150 mM NaCl and 50 mM TrisHCl; pH 7.6, TBS). The quantity of DNA was measured with a Thermo Scientific Nanodrop spectrophotometer; then, the samples were normalized with TBS buffer. The small volumes of 5 μL of solubilized DPCC isolates were transferred to a nitrocellulose membrane as dots. After the samples were dried, the blocking procedure with bovine serum albumin (BSA, 3% *w/v* in phosphate-buffered saline with 0.05% Tween-20 (PBST)) was followed for 1 h at room temperature. After blocking, the membrane was washed with PBST 3 times for 5 min on a shaker, then incubated with 10 mL of the primary antibody detecting Top1 (Invitrogen, Waltham, MA, USA) (1:1000 in 1.5% BSA in PBST) overnight at 4 °C on a shaker. After incubation, the membrane was washed with PBST 3 times for 5 min on a shaker. Then, 10 mL of the secondary antibody, Peroxidase-AffiniPure Goat Anti-Rabbit IgG (H+L) (Cat#111-035-003, Jackson Immuno Research Inc, Baltimore Park, PA, USA) (1:3000 in 1.5% BSA in PBST), was added and incubated for 1 h at room temperature on a shaker. Before the detection, the membrane was washed with PBST 5 times for 5 min on a shaker, after which 1 mL of Immobilon Forte Western HRP Substrate (Cat# WBLUF0500, Millipore, Darmstadt, Germany) was added and incubated with shaking for 1 min. The detection was performed with a BIORAD Protein Detection System. Quantification was performed on Fiji image-J. Statistics calculation were calculated with GraphPad Prism version 9.0.0 (GraphPad Prism, RRID:SCR_002798), California, USA. The significance of differences was determined using unpaired Welch’s correction and two-tailed *t*-tests (“ns” < 0.1234, * *p* < 0.0332, ** *p* < 0.0021, *** *p* < 0.0002, and **** *p* < 0.0001).

### 4.19. Immunoprecipitation and Immunoblotting

Cells were lysed with lysis buffer (50 mM Tris-HCl (pH 7.4), 150 mM NaCl, 1% Triton X-100, 5 mM EDTA, 1 mM Na_3_VO_4_, 50 mM β-glycerophosphate, and 50 mM NaF) supplemented with Protease Inhibitor Cocktail (Cat. #P8340, Sigma) and phenylmethylsulfonyl fluoride (PMSF). Samples were adjusted to have an equal concentration of total protein and subjected to PAGE electrophoresis followed by immunoblotting with the primary antibody (Phospho-Histone H2A.X (Ser139) Antibody, Cell Signaling Technology Cat# 2577, RRID:AB_2118010) [71].

### 4.20. Hypergeometric Distribution

To predict the significance of the overlap between triple mutations and Top1 cleavage sites, we used R language tools (see code availability section for applicable codes). The problem of overlap at DNA sites is described by a hypergeometric distribution, where one list defines the number of cleavage sites, and the other list defines the number of mutations. Assume the total genome size is *n*, the number of points in the first list is *a,* and the number of points in the second list is *b*. If the intersection between the two lists is *t*, the probability density of seeing *t* can be calculated as *Probability of occurrence = dhyper (t, a, n–a, b).* Based on the existing data, *b* < *a*. Thus, the largest possible value for *t* is *b*. Therefore, the p-value of seeing intersection *t* is *sum (dhyper (t:b, a, n–a, b))*. Taking the top1 cleavage sites, (a) we cover 2.5 × 10^7^ base pairs in the genome (2 × 10^5^ sites × 50 bp per peak); the number of mutations (b) = 46,099, the total genome (n) = 3 × 10^9^ bp, and t = 9000, representing the overlap out of these 46,099 mutations. The sum of the overlap was computed to be absolute zero with a null probability.

## 5. Conclusions

Here, we uncovered a novel mechanism for the development of irinotecan resistance in colon cancer cells. We established that:Top1 creates cleavage sites at specific sites in DNA, mostly in the satellite regions.Due to the evolution of cancer, cancer cells have a higher number of such sites.The repair of Top1-generated DSB upon irinotecan treatment generates mutations at the cleavage sites, which prevent interactions with Top1 upon further drug exposures.

The accumulation of such mutations at Top1 sites over multiple drug exposures leads to a reduction in Top1 binding to DNA and the inability to generate toxic DSB upon irinotecan treatment.

## Figures and Tables

**Figure 1 ijms-24-08717-f001:**
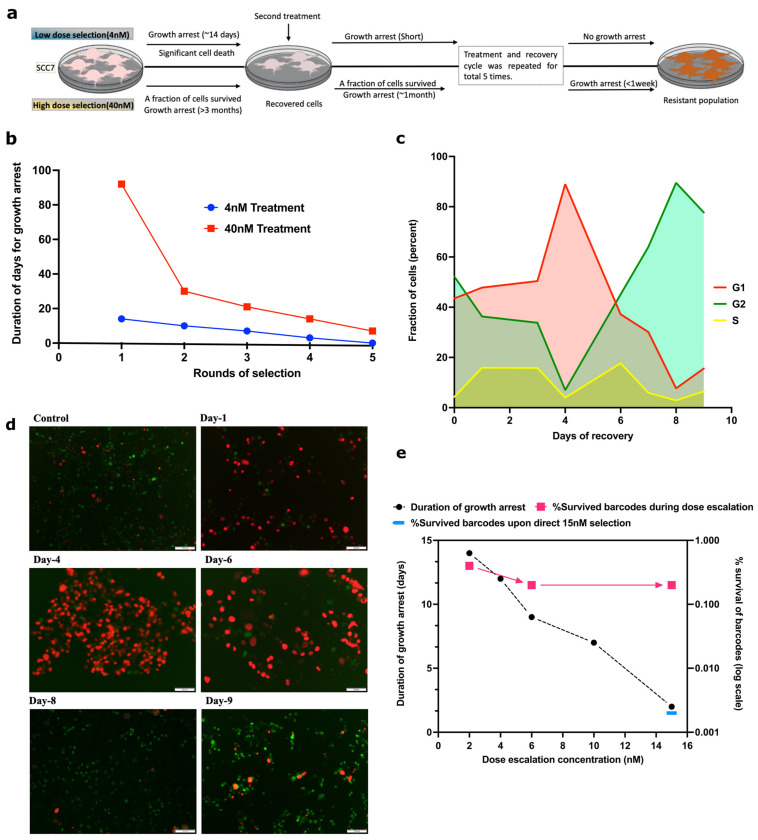
Adaptation to SN-38 is associated with reduction in duration of growth arrest. (**a**) Graphical sketch of the experiment plan, showing low-dose selection (upper panel) and high-dose selection (lower panel). (**b**) Clones of HCT116 cells were exposed to the same dose of SN-38 (either 4 nM or 40 nM for 24 h) multiple times, and periods of the cell cycle arrest following exposures were measured. (**c**,**d**) Simultaneous recovery of SN-38-treated cells from cell cycle arrest. Cells were infected with cell cycle reporter plasmid (see Section 4). Cells were treated with 4 nM SN-38 and, following the cell cycle arrest (Days 1 and 4), allowed to recover (Day 8). Imaging was performed on the indicated days following SN-38 exposure using TexRed (red) and FITC (green) channels. Green fluorescence represents G2, red fluorescence represents G1, and overlapping of red and green shows S phase. Images are at scale of 100 µm in Figure 1**d**. Quantification of data for fraction of G1, G2 and S phase cells during recovery from 4 nM SN-38 treatment was quantified by imageJ. Experiments were carried out in triplicate wells for each treatment. (**e**) Cells were exposed to rounds of treatments with 2, 4, 6, 10, and 15 nM SN-38, and the periods of recovery were measured. Survival of barcodes after these treatments is shown on the same graph. Cells were barcoded using the 50 million lentiviral barcoding libraries (Cellecta), and cells recovered after the treatments were collected. Barcodes were isolated, sequenced by IonTorrent, and analyzed. Quantification of barcode survival is represented on right Y axis. A total of 0.4% of total barcodes survived 2 nM dose compared to untreated control (100%). Following further treatment with 6 nM and 15 nM, 0.2% of initial barcodes survived, i.e., approximately 50% of barcodes that survived the 2 nM dose. Also shown is the fraction of barcodes that survived 15 nM dose without preadaptation to lower doses (0.002%).

**Figure 2 ijms-24-08717-f002:**
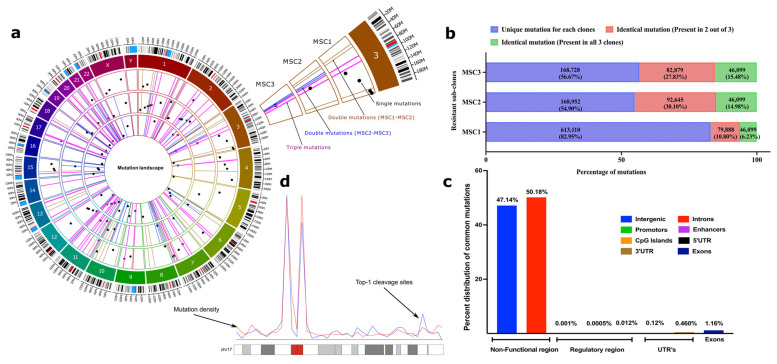
SN-38-resistant subclones harbor identical mutations in their genomes in non-coding regions. (**a**) Circos plot to represent the category of mutations based on their occurrence in three independent clones. Three circles inside the chromatogram notations represent three individual selected resistant clones (MSC1, MSC2, and MSC3 from outside to inside). Dots (black) inside the respective circles represent mutations that are unique in the clones and not overlapping with others. Bars (brown) represent the overlapping mutations in clones MSC1 and MSC2. Bars (blue) represent the overlapping mutations in clones MSC2 and MSC3. Bars (pink) represent mutations that are overlapping in all three clones. Only 50 mutations of each category were chosen randomly and plotted for visualization purposes to their genomic locations. The plot was prepared using a commercial license from OMGenomics (Circa, RRID:SCR_021828) application. (**b**) The total quantification of overlapping mutations in the SN-38-resistant clones that emerge in the adaptation process. Stack graph showing the overlapping mutations among resistant sub-clones. Mutations in the resistant clones that differ from SCC1 were compared to each other. These mutations represent three categories: (blue) mutations that are unique to each clone and not found in other clones, (red) mutations that are identical in two clones, and (green) mutations that are identical in all three clones. (**c**) Categorical distribution of mutations to the non-functional, regulatory and UTR region of genome computed from annotation using SnpEff. (**d**) An example representation of chromosome 17 showing overlapped mutation density (blue line) and Top-1 cleavage sites (red line).

**Figure 3 ijms-24-08717-f003:**
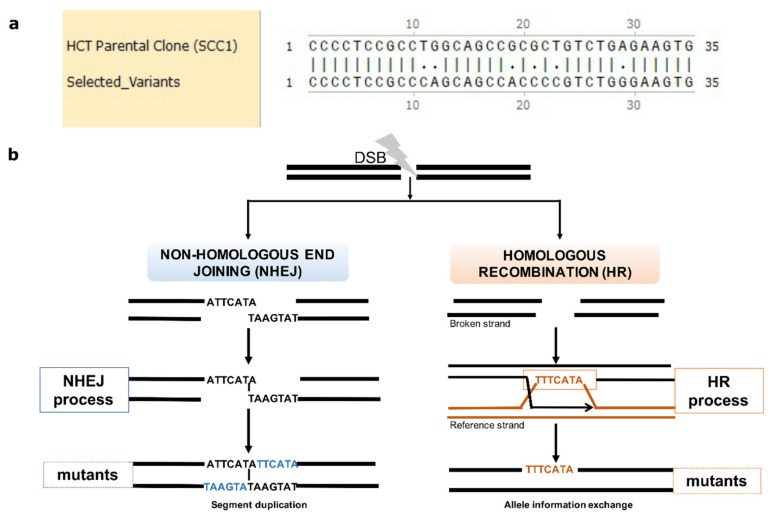
Mutations emerge in clustering patterns resulting from HR and NHEJ repair. (**a**) Example for clustering pattern of mutations compared to parental SCC1. (**b**) Sketch diagram of the mechanisms of DSB repair that leave specific mutation signatures. Real mutations found in all three resistant clones that result from NHEJ and HR repair are shown as examples. During NHEJ repair, both DNA ends are either trimmed, which results in deletions, or alternatively are paired via short homology region, which results in duplication of strands on both strands (insertions) (left panel). In case of HR, an unbroken allele at the homologous chromosome can serve as template that is copied to the broken strand. Sister chromatids can also be used as templates, but we do not see such events since they do not generate mutations.

**Figure 4 ijms-24-08717-f004:**
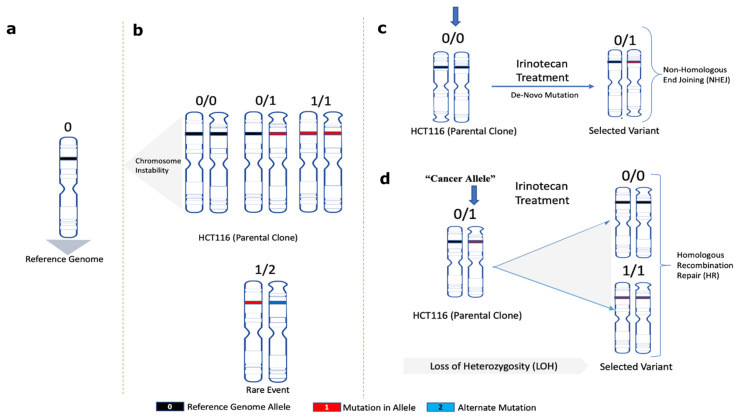
A framework model for analyses of the role of mutations in adaptation to SN-38. Schematic representation for (**a**) normal reference genome allele, (**b**) alleles found in parental SCC1 clone, (**c**) de novo mutations in MSC clones after NHEJ repair, (**d**) allele change upon loss of heterozygosity after HR repair (loss of heterozygosity herein is referred to as the change of mutated heterozygous alleles reversed to homozygous reference in selected variants).

**Figure 5 ijms-24-08717-f005:**
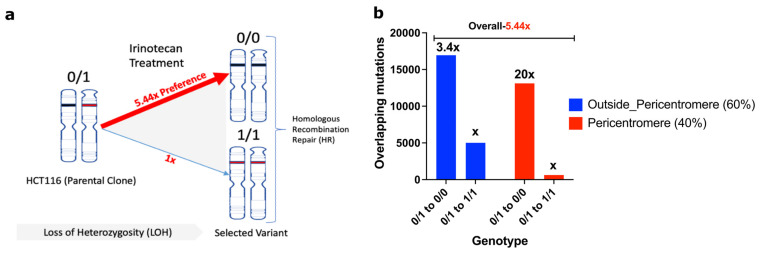
Cancer alleles are more susceptible to breaks compared to reference genome. (**a**) HR-repair-generated mutations revert to the reference genome allele 5.44 times more often than to the cancer allele. Therefore, cancer alleles are broken by Top1 5.44 times more often than reference genome alleles. Accordingly, reverting to the reference genome alleles protects from DSBs upon consequent exposures to SN-38. (**b**) Prevalence of breaks in cancer alleles compared to reference genome alleles is lower in chromosome arms than in the pericentromeric region.

**Figure 6 ijms-24-08717-f006:**
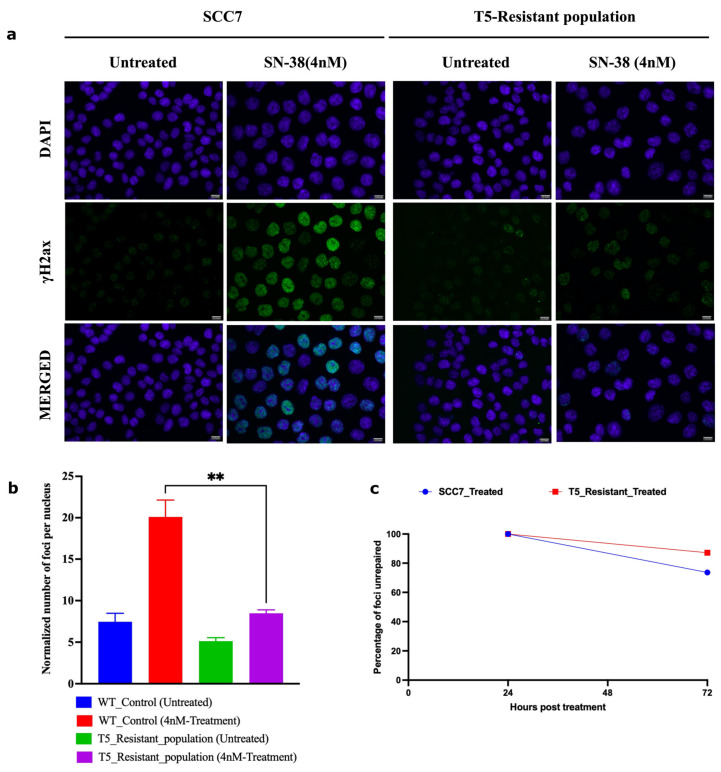
Adapted population experiences lower frequency of DSBs following SN-38 exposure. (**a**) γH2AX foci in cells exposed to SN-38 (4 nM for 24 h). SCC7 cells are compared to the same clone after five cycles of adaptation to 4 nM SN-38. Experiment was conducted in biological triplicates. (**b**) Quantification of data presented in (**a**) of the number of foci generated 24 h post-treatment; *n* = 368 images were analyzed with the integrated software (refer to Section 4), images are at scale of 12.5 µm. (**c**) Line plot shows the number of foci remaining after 72 h of recovery from SN-38 (SCC7, 73%; T5_resistant, 87%), indicating that the rate of DSB repair is not faster in adapted cells. Data representing *n* = 431 images were analyzed using integrated software (refer to Section 4). Statistics were calculated using GraphPad Prism version 9.0.0, California, USA. The significance of differences was determined using an unpaired Welch’s correction two-tailed *t*-test (** *p* < 0.0021) denoted above in (**b**).

**Figure 7 ijms-24-08717-f007:**
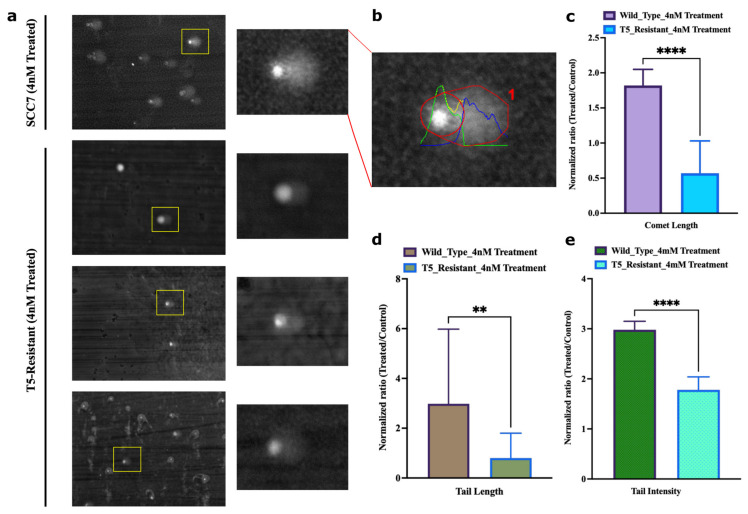
Adapted population experiences lower frequency of overall breaks following SN-38 exposure. (**a**) Representative image of the field of SCC7 cells treated with SN-38 (24 h) and corresponding comets, imaged at 20× magnification, scale at 12.5 µm. Insert shows enlarged image of the comet. Lower three images with inserts show representative images of the adapted population. (**b**) Representative of quantified output image used to compute parameters for comet analysis. (**c**) Comet length and (**d**) comet tail length. (**e**) Intensity of DNA in tail was estimated using OpenComet v1.3.1. Bar plot represents normalized ratio of treated and control comets in SCC7 and T5-adapted cells separately. Statistics were calculated as mean of *n* = 21 comets in each using GraphPad Prism version 9.0.0, California, USA. The significance of differences was determined using unpaired Welch’s correction and two-tailed *t*-test (** *p* < 0.0021, **** *p* < 0.0001), as denoted in above figures (**c**–**e**).

**Figure 8 ijms-24-08717-f008:**
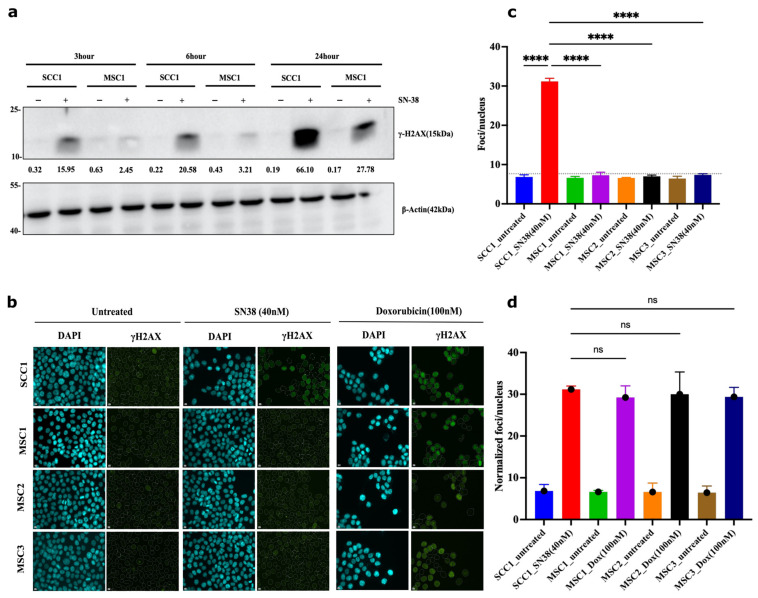
Mutants experience lower frequency of DSB following SN38 exposure. (**a**) γH2AX foci in cells exposed to SN38 (40 nM for 30 min). After 3, 6, and 24 h, cells were harvested and lysed for immunoblot for quantification of phosphorylated H2AX levels, as shown. Experiment was conducted in biological duplicates (*n* = 2). (**b**) SCC1 cells are compared to the mutant clones using γH2AX foci quantification for 24 h treatment observing significant differences based on immunoblot phosphor-H2AX levels. We observe consistent increases in protein levels with highest expression for SCC1 at 24 h. Experiment was conducted in biological triplicates (*n* = 3). (**c**) Quantification of SN-38-generated foci presented in (**b**) showing significant decrease in foci generation in resistant mutants. *n* = 228 images were analyzed with the integrated software (refer to Section 4); for short-term treatment (30 min data) and foci quantification, refer to Appendix A. Quantification for doxorubicin cell sensitivity is presented in Appendix A. Statistics were calculated using GraphPad Prism version 9.0.0, California, USA. The significance of differences was determined using unpaired Welch’s correction and two-tailed *t*-test (ns < 0.1234, **** *p* < 0.0001), as denoted in above in figure (**c**). Pseudo-color is used in DAPI for visualization purposes. Controls in (**c**,**d**) are the same.

**Figure 9 ijms-24-08717-f009:**
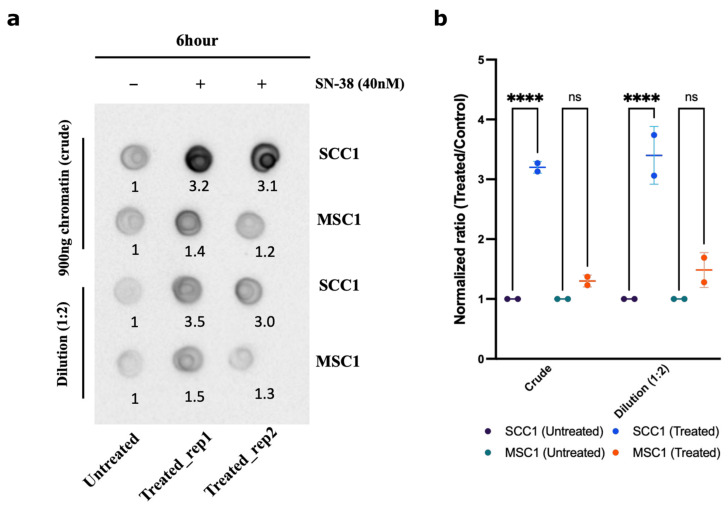
Chromatin fractionation and covalently bound Top1. Parental line SCC1 and resistant MSC1 were treated with 40 nM of SN-38 for 6 h, and chromatin was isolated. Top1 covalently bound to chromatin was quantified using dot blot with Top1 antibody. (**a**) Dot blot showing the crude extract upper panel and dilution (1:2) in lower panel (for raw blot, refer to raw images). (**b**) Statistics were calculated for (*n* = 2) independent experiments using GraphPad Prism version 9.0.0, California, USA. The significance of differences was determined using unpaired Welch’s correction and two-tailed *t*-test (ns < 0.1234, **** *p* < 0.0001), as denoted.

## Data Availability

Data such as raw FASTQ files for human whole-genome sequencing have been submitted to the publicly accessible database SRA under accession number PRJNA738674. Processed VCF showing identical mutations among all resistant subclones compared to the parental line (available in Appendix A) is included in this paper. A full list of genes from transcriptome analysis and shRNA screens are available in this paper. Raw data and files after the transcriptome analysis are deposited and are available in the GEO database with accession number GSE189366. The analyzed data are presented in Appendix A in the relevant sections. Already published elsewhere used in analysis: These data were referred to from literature: GSE57628, mapping of Top-1 binding and cleavage sites reported for HCT116 cells; summary of epigenome ENCSR309SGV (ENCODE database that presents a summary of various methylation signatures, such as H3K9me3 and H3K27me3, for the genome of HCT116). All the codes essentially used in the analysis of sequencing data, either human whole-genome, transcriptome analysis, or shRNA screening, have been submitted to the public domain and can be accessed through Github (https://github.com/santoshbiowarrior333/Irinotecan_resistance).

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
