# Peer review of "Evolution of Resistance to Irinotecan in Cancer Cells Involves Generation of Topoisomerase-Guided Mutations in Non-Coding Genome That Reduce the Chances of DNA Breaks"

_ijms, 2023, doi:10.3390/ijms24108717_

Round 1

Reviewer 1 Report

The manuscript “Evolution of resistance to Irinotecan in cancer cells involves generation of topoisomerase-guided mutations in non-coding genome that reduce the chances of DNA breaks” by S. Kumar et al. is devoted to the study of the mechanism of development of tumor cells resistance to chemotherapy. The resistance of tumors to treatment remains an acute problem despite all the successes in the development of modern methods of cancer therapy. The presented manuscript shows that resistance does not arise as a result of pre-existing mutations or phenotypic differences in resistant cells, but develops in the course of treatment. A new unexpected mechanism of resistance has been proposed - accumulation of recurrent mutations in non-coding regions of DNA at Top1-cleavage sites, which removes tumor cells from the action of the drug. The article is made using modern methods, well written and can be published in its present form.

Author Response

Reviewer 1 did not have any concerns.

Reviewer 2 Report

Characterization and study of mechanism of drug-resistance under the dose-escalation is important for the understanding of cancer drug resistance.  This manuscript describes an interesting mechanism of irinotecan resistance involving mutation of Top1-dependent DNA breaking site in non-coding regions.  The quality of the data is generally very good, and supports the conclusion.  Here are my comments that should be addressed.

1. The authors should mention how the Top1 cleavage sites mapped in previous studies are distributed relatively between transcribed and untranscribed regions.  Due to the well known role of Top1 in transcription, one would expect a majority of Top1 cleavage sites be found near or in transcribed genes.  The authors report here that “mutation sites strongly co-localized with sites of DNA cleavage by Top1 [55] (19.8% of cases)”.  However the mutation sites are reported to be localized in repeats and untranslated regions.  Is there a reason why the fraction of Top1 cleavage sites in transcribed regions do not lead to mutations associated with SN-38 resistance?

2. The authors proposed that the mutations in SN-38 resistant clones result in DNA structure that make it less prone to Top1-induced breaks. They should discuss what is known in the literature on Top1 sequence binding preference, and whether it corelates with the Top1 preferred cleavage sites in the cancer alleles as well as the Top1 cleavage resistant mutations identified in this study. 

Minor criticism:

The figure panels are labeled in lower case letters, but are referred to with upper case letters in the text.

Author Response

Reviewer 2 asked (1) to explain why most of the Top1 cleavage sites is located in silenced regions, while Top1 has been implicated in transcription, and (2) discuss binding sites for Top1 in Discussion.

We introduced in the Discussion a paragraph explaining that Top1 binds relatively non-specifically, but cleaves at specific sites. These site were shown to locate both transcribed and silenced regions, including centromere. Accordingly, it appears that beside transcription Top1 may serve in relieve of supercoils in other processes, e.g. replication or repair.